# Toward Understanding Privileged Features Distillation in Learning-to-Rank

**Shuo Yang**\*
UT Austin
yangshuo_ut@utexas.edu

**Sujay Sanghavi**
Amazon
sujayrs@amazon.com

**Holakou Rahmanian**
Amazon
holakou@amazon.com

**Jan Bakus**
Amazon
jbakus@amazon.com

**S.V.N. Vishwanathan**
Amazon
vishy@amazon.com

## Abstract

In learning-to-rank problems, a *privileged feature* is one that is available during model training, but not available at test time. Such features naturally arise in merchandised recommendation systems; for instance, "user clicked this item" as a feature is predictive of "user purchased this item" in the offline data, but is clearly not available during online serving. Another source of privileged features is those that are too expensive to compute online but feasible to be added offline. *Privileged features distillation* (PFD) refers to a natural idea: train a "teacher" model using all features (including privileged ones) and then use it to train a "student" model that does not use the privileged features.

In this paper, we first study PFD empirically on three public ranking datasets and an industrial-scale ranking problem derived from Amazon's logs. We show that PFD outperforms several baselines (no-distillation, pretraining-finetuning, self-distillation, and generalized distillation) on all these datasets. Next, we analyze why and when PFD performs well via both empirical ablation studies and theoretical analysis for linear models. Both investigations uncover an interesting non-monotone behavior: as the predictive power of a privileged feature increases, the performance of the resulting student model initially increases but then decreases. We show the reason for the later decreasing performance is that a very predictive privileged teacher produces predictions with high variance, which lead to high variance student estimates and inferior testing performance.

## 1 Introduction

For recommendation systems, the features at test time are typically a subset of features available during training. Those missing features at test time are either too expensive to compute in real-time, or they are post-event features. For instance, for an e-commerce website, "click" is a strong feature for predicting "purchase", but "click" exists as a feature only in the offline training data, but not during online serving (i.e., one cannot observe "click" before recommendations are generated). Those features that exist only during training are called *privileged features*. Those that exist during both training and testing are called *regular features* [XLG+20].

The naive approach is to ignore the privileged features and train a model that only takes regular features. Such methods inevitably miss the information in the privileged features and lead to inferior

---

\*This work was done while Shuo Yang was interning at Amazon.

36th Conference on Neural Information Processing Systems (NeurIPS 2022).

performance. A natural instinct to resolve this is to *(a)* use the privileged features (either by themselves [LPSBV16] or in conjunction with regular features [XLG+20]) to train a "teacher" model, and then *(b)* use it to transfer information via distillation[2] into a "student" model that only uses the regular features. The approach of a teacher only using privileged features is named *generalized distillation (GenD)* [LPSBV16], and the approach of a teacher using both privileged and regular features has been referred to as *privileged feature distillation (PFD)* [XLG+20].

**In this paper** we provide a detailed investigation – first via empirical ablation studies on moderate-scale public and industrial-scale proprietary datasets with deep-learning-to-rank models, and second via rigorous theoretical analysis on simple linear models – into why and when privileged feature distillation works and when it does not. While this paper focuses on learning-to-rank, our results apply to regression/classification problems in general. As a summary, our **main contributions** are:

- We evaluate PFD on three moderate-scale public ranking datasets: Yahoo, Istella, and MSLR-Web30k, and an industrial-scale proprietary dataset derived from Amazon search logs.

- In all evaluated settings, PFD is better than or as good as the baselines: no-distillation, GenD (teacher model only uses privileged features), self-distillation (teacher model only uses regular features), and pretraining on privileged features then finetuning (when applicable) (Table 2).

- We conduct comprehensive ablation studies for PFD. We find that
  - PFD is effective as long as the teacher loss dominates the distillation loss and the performance is not sensitive to $\alpha$. Specifically, distillation loss is a linear combination of the loss w.r.t. data and the loss w.r.t. teacher predictions and $\alpha$ is the mixing ratio (Figure 3).
  - While it is known that the gains from self-distillation (over a no-distillation one-shot training baseline) are larger when the positive labels are sparser, we see that these gains are *further* amplified by PFD; i.e. the relative gain of PFD over self-distillation also increases as the labels become sparser (Figure 4).
  - Non-monotonicity in the effectiveness of PFD: as the predictive power of a privileged feature increases, the resulting student performance initially increases but then decreases (Figure 5).

- To provide a deeper insight into the landscape of privileged features and distillation, we next rigorously analyze it in a stylized setting involving linear models. We show that
  - PFD works because the teacher can explain away the variance arising from the privileged features, thus allowing the student to focus on the part it can predict. (Theorem 1).
  - The reason that GenD is inferior to PFD (as seen in our empirical evaluation) is because it results in a weaker teacher, and also because in the case where the privileged and regular features are independent, the teacher predictions appear as pure noise to the student (who cannot learn from them) (Remark 2).
  - A very predictive privileged feature induces high variance teacher predictions, which lead to inaccurate student estimates and inferior testing performance. This explains the observation that the most predictive privileged features do not give the best performance (i.e., the non-monotonicity) in our empirical ablation studies (Theorem 2).

The rest of the paper is organized as follows: Section 2 covers related works. Section 3 introduces the problem setup, the PFD algorithm and other algorithms for comparison. Section 4 presents empirical evaluation and ablation studies of PFD; and Section 5 presents theoretical insights.

## 2 Related Work

**Privileged features** widely exist in different machine learning problems, including speech recognition [MM16], medical imaging [GCA+19], image super-resolution [LLKH20], etc [FA12, FTRS13, FKSH14, ALL17]. Privileged features are not accessible during testing either because they are too expensive to compute in real time, or because they are post-event features (thus cannot be used as input) [CM18].

**Learning with privileged features** is pioneered in [VV09], where they propose a framework named "learning using privileged information" (LUPI). At the core, LUPI uses privileged information to

---

[2] Here, by distillation we mean the standard practice of labeling the training dataset using teacher predictions, and using these as supervision targets in the training of the student model.

distinguish between easy and hard examples. The methods are thus closely related to SVM, as the hardness of an example can be expressed by the slack variable. For instance, [VV09, PIVV10] propose the "SVM+" algorithm which generates slack variables from privileged features and learns an SVM based on regular features with those slack variables; [SQL13] proposes a pair-wise SVM algorithm for ranking, which uses privileged features to distinguish easy and hard pairs. [LHS14] presents a variation where the privileged features are used to generate importance weighting for different training samples. Empirically, [SER14] demonstrates that whether LUPI is effective critically depends on experimental settings (e.g., preprocessing, training/validation split, etc). [VI15] considers transferring the kernel function from a teacher SVM that only uses privileged features to a student SVM that only uses regular features; [LDX$^+$20] extends the SVM+ algorithm to imperfect privileged features.

**Model distillation** [HVD$^+$15] is a common method for knowledge transfer, typically from a large model to a smaller one [PPA18, GYMT21]. Recent works have shown great empirical success in ranking problems [TW18, HAS$^+$20, RPM$^+$21] and even the cases where the teacher model and student model have the identical structure [FLT$^+$18, QYT$^+$21].

**Using distillation to learn from privileged features** are first proposed in [LPSBV16] as "generalized distillation" (GenD). It provides a unified view of LUPI and distillation. GenD, along with the variants [MM16, GMM19, LLKH20], train a teacher model with only privileged features and then train a student model to mimic the teacher's predictions. PFD is recently proposed in [XLG$^+$20], where the teacher model takes both regular and privileged features as input. PFD and GenD differ from the standard model distillation as they focus on exploiting privileged features but not on reducing the model size. [XLG$^+$20] empirically demonstrates the superior performance of PFD for recommendation systems on a non-public data set.

**Understanding of privileged features distillation** is lacking, despite the aforementioned empirical success. Previously, [PV10] shows that LUPI brings faster convergence under a strong assumption that the best classifier is realizable with only privileged features. [LPSBV16] shows that GenD enjoys a fast convergence rate. It assumes that the teacher model has a much smaller function class complexity than the student model, which does not match with PFD. [GCFY18] studies GenD under semi-supervised learning and shows that the benefits come from student function class complexity reduction. However, it does not quantify such reduction and the theory does not explain what is the benefit of using privileged features. To the best of our knowledge, there is no empirical or theoretical study explaining why PFD is effective.

**Other ways of utilizing privileged features** are also previously proposed. [CJFY17] uses privileged information to learn a more diverse representation to improve image classification performance. [LLKH20, WZW$^+$21] propose distillation schemes for better feature extraction from regular features. A more recent work [CJKB22] considers training a model with both regular and privileged features to obtain a better internal representation of the regular features.

## 3 Problem Setup and Algorithms

Consider a learning-to-rank problem where each query-document pair has features $\mathbf{x} \in \mathcal{X}$ and $\mathbf{z} \in \mathcal{Z}$ and a label $y \in \mathcal{Y}$ (e.g., click or human-annotated relevance) drawn from an unknown distribution $\mathcal{D}(y|\mathbf{x}, \mathbf{z})$. Suppose $\mathbf{x}$ is the regular feature that is available during both training and testing and $\mathbf{z}$ is only available during training. Concretely, *privileged feature* is defined in the literature as below:

**Definition 1** (Privileged Feature [CJKB22]). *For feature $\mathbf{z}$ that exists during training but not testing, we say $\mathbf{z}$ is a **privileged feature** if and only if $I(y; \mathbf{z}|\mathbf{x}) := H(y|\mathbf{x}) - H(y|\mathbf{x}, \mathbf{z}) > 0$.*

Conditional mutual information $I(y; \mathbf{z}|\mathbf{x})$ and conditional entropy $H(\cdot|\cdot)$ follow from the standard notation of information theory. According to Definition 1, the privileged feature $\mathbf{z}$ provides extra predictive power of $y$. For the rest of this paper, we focus on the setting that $\mathbf{z}$ is a privileged feature.

**Remark 1.** *An implication of Definition 1 is that the privileged feature $\mathbf{z}$ can be independent of the regular feature $\mathbf{x}$. In such cases, any transformation of $\mathbf{z}$ is not learnable from $\mathbf{x}$, and therefore using $\mathbf{z}$ as auxiliary learning target does not help. Interestingly, PFD can still improve the student performance, even when $\mathbf{z}$ and $\mathbf{x}$ are independent (see Section 5).*

We consider the following general learning problem: we are given a *labeled training set* of size $n$, $\mathcal{S}_{label} := \{(\mathbf{x}_i, \mathbf{z}_i, y_i)\}_{i \in [n]}$, and a *unlabeled training set* of size $m$, $\mathcal{S}_{unlabel} := \{(\mathbf{x}_i, \mathbf{z}_i)\}_{i \in [m]}$. Our

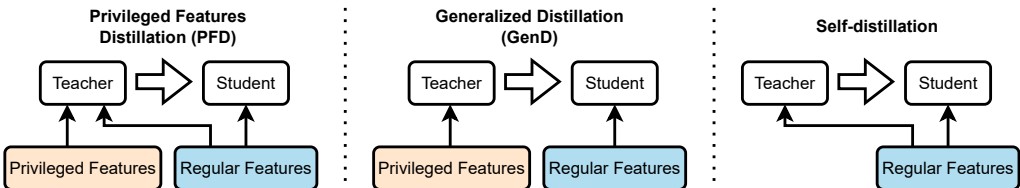

Figure 1: Illustration of PFD, generalized distillation (GenD) and self-distillation.

goal is to generate good ranking based only on regular features $\mathbf{x}$. For clarity of exposition, we only consider pointwise scoring functions $\mathcal{F} := \{f \mid f : \mathcal{X} \mapsto \mathcal{Y}\}$, which generates a score for each document, and the ranking is induced by sorting the scores. The results in this paper can be easily extended to models beyond pointwise scoring functions (e.g., DASALC [QYZ$^+$21]).

The distinction between labeled and unlabeled datasets is for generality. The unlabeled dataset naturally appears in recommendation systems, where the majority of search logs do not contain any user interactions. Instead of taking all such logs as negative samples, it is more proper to view them as unlabeled data due to the lack of user engagement. For the logs that contain click, the documents therein with no click can be treated as negative samples.

### 3.1 Privileged features distillation

PFD first trains a teacher model that takes both $\mathbf{x}$ and $\mathbf{z}$ as input to predict $y$, i.e., teacher function class is $\mathcal{G}_{PFD} := \{g \mid g : \mathcal{X} \times \mathcal{Z} \mapsto \mathcal{Y}\}$. For simplicity, we consider pointwise loss $l : \mathcal{Y} \times \mathcal{Y} \mapsto \mathbb{R}$ in this section, while the method can be easily extended to other loss functions (see an extension to pairwise loss in Section 4). The privileged features distillation takes the following two steps:

**Step I: Training a teacher model** $g_{PFD} \in \mathcal{G}_{PFD}$ by minimizing the loss on the labeled dataset: $\sum_{(\mathbf{x}_i, \mathbf{z}_i, y_i) \in \mathcal{S}_{label}} l\left(g(\mathbf{x}_i, \mathbf{z}_i), y_i\right)$. In practice, gradient-based optimizer is used for loss minimization.

**Step II: Training a student model by distillation.** The teacher model $g_{PFD}$ trained from *Step I* is used to generate pseudo labels on $\mathcal{S}_{label}$ and $\mathcal{S}_{unlabel}$. Let $\mathcal{S}_{all}$ denote the union of $\mathcal{S}_{label}$ and $\mathcal{S}_{unlabel}$. The student model is trained by minimizing the following *distillation loss*:

$$\alpha \cdot \underbrace{\sum_{(\mathbf{x}_i, y_i) \in \mathcal{S}_{label}} l(f(\mathbf{x}_i), y_i)}_{\textit{data loss}} + (1 - \alpha) \cdot \underbrace{\sum_{(\mathbf{x}_i, \mathbf{z}_i) \in \mathcal{S}_{all}} l\left(f(\mathbf{x}_i), g_{PFD}(\mathbf{x}_i, \mathbf{z}_i)\right)}_{\textit{teacher loss}}, \tag{1}$$

where $\alpha \in (0, 1)$ controls the mixing ratio between the data loss and teacher loss. The student model is trained by minimizing the distillation loss in Equation (1).

### 3.2 Other algorithms for comparisons

Here we introduce two other algorithms for comparison. See illustration in Figure 1.

**GenD** [LPSBV16] is a distillation method where the teacher model takes only privileged features as input, i.e., the teacher function class is $\mathcal{G}_{GenD} = \{g \mid g : \mathcal{Z} \mapsto \mathcal{Y}\}$. The teacher model $g_{GenD} \in \mathcal{G}_{GenD}$ is obtained by minimizing $\sum_{(\mathbf{z}_i, y_i) \in \mathcal{S}_{label}} l\left(g(\mathbf{z}_i), y_i\right)$. Similar to PFD, the distillation loss is a linear combination of the data loss and teacher loss.

**Self-distillation** [FLT$^+$18, QYT$^+$21] is a distillation method where the teacher model has the same structure as the student model. Specifically, the teacher model $g_{self\text{-}dist.} \in \mathcal{F}$ is obtained by minimizing $\sum_{(\mathbf{x}_i, y_i) \in \mathcal{S}_{label}} l\left(g(\mathbf{x}_i), y_i\right)$. Notice that $\mathcal{F}$ is also the student function class. Similar to PFD, the distillation loss is a linear combination of the data loss and teacher loss. Comparing PFD against self-distillation separates the benefits of adopting privileged features and distillation.

## 4 Experiments

### 4.1 Main results on public datasets

We first evaluate the performance of PFD on three widely used public ranking datasets. Specifically, we use the Set1 from "Yahoo! Learn to rank challenge" [CC11]; "Istella Learning to Rank" dataset

[DLN$^+$16]; and Microsoft Learning to Rank "MSLR-Web30k" dataset [QL13]. We refer to them as "Yahoo", "Istella" and "Web30k" throughout this section.

**Datasets overview and preprocessing.** The training samples in all three datasets can be viewed as *query groups*, where each query group contains 1 query and multiple documents to be ranked. Each query-document pair is represented as a real-value feature vector (e.g., user dwelling time, tf-idf of document, etc. See [CC11] for detail). Further, each query-document pair has a human-annotated relevance score $r \in \{0, 1, 2, 3, 4\}$. All datasets are preprocessed by removing query groups that contain no positive relevance score or have less than 10 documents. The features are transformed by the log1p transformation as in [ZWBN20, QYZ$^+$21].

**Binary label generation.** In practice, binary label (e.g., click) is more commonly seen and easier to obtain than relevance score. For our experiments, we generate a binary label $y$ for each query-document pair based on the human-annotated relevance score $r$. Specifically:

$$y = \mathbb{I}\left(t \cdot r + G_1 > t \cdot \tau_{\text{target}} + G_0\right), \tag{2}$$

where $t$ is a temperature parameter and $G_1$ and $G_0$ follow the standard Gumbel distribution. It can be shown that $y$ is 1 with probability $\sigma(t \cdot (r - \tau_{\text{target}}))$, where $\sigma(\cdot)$ is the sigmoid function (see Appendix A.1 for proof). For the rest of our experiments, we set $t = 4$ and $\tau_{\text{target}} = 4.8$ unless otherwise mentioned. We refer to the query groups that contain at least one $y = 1$ to be *positive query groups*, and other query groups are referred to as *negative query groups*.

**Regular and privileged features split.** For each of the datasets, we sort the features according to the magnitude of their correlations with the binary label $y$ and use the top 200, 50, and 40 features as privileged features for Yahoo, Istella, and Web30k, respectively. Other features are used as regular features. Please see Table 1 for dataset statistics after preprocessing and binary label generation.

| Data set | # features | | # query groups | | # docs per query group | | # positive query group | |
|---|---|---|---|---|---|---|---|---|
| | regular | privileged | training | test | training | test | training | test |
| Yahoo | 500 | 200 | 14,477 | 4,089 | 30.02 | 29.93 | 2.46% | 2.08% |
| Istella | 170 | 50 | 23,171 | 9,782 | 315.90 | 319.62 | 8.83% | 9.13% |
| Web30k | 96 | 40 | 18,151 | 6,072 | 124.35 | 123.34 | 3.54% | 3.49% |

Table 1: **Dataset statistics** after preprocessing and binary label generation (Equation (2)). Web30k statistics are calculated based on the fold-1 of the official 5 fold splits. Recall that "positive query groups" are query groups that contain at least 1 document with $y = 1$.

**Ranking model and performance metric.** The ranking model is a 5-layer fully connected neural network, which maps the query-document feature into a real-value score $s \in [0, 1]$. The ranking $\widehat{\pi}$ of documents is obtained by sorting the scores decreasingly, where $\widehat{\pi}(i)$ represents the ranked order of the $i$-th document. The ranking performance is measured by the NDCG@$k$ metric:

$$\text{NDCG@}k(\widehat{\pi}, \mathbf{y}) = \frac{\text{DCG@}k(\widehat{\pi}, \mathbf{y})}{\text{DCG@}k(\pi^*, \mathbf{y})}, \quad \text{DCG@}k(\pi, \mathbf{y}) = \sum_{\pi(i) \leq k} \frac{2^{y_i} - 1}{\log_2(1 + \pi(i))},$$

where $\pi^*$ is the optimal ranking obtained by sorting $y_i$.

**PFD is effective for all three datasets.** We evaluate the efficacy of PFD on all three aforementioned datasets, under both pointwise (RankBCE) and pairwise (RankNet [BSR$^+$05]) loss functions (see definitions in Appendix A.2). Please see the evaluated algorithms and results in Table 2 (complete results with RankNet loss deferred to Table 4). Figure 2 shows the testing NDCG@8 curve on Yahoo and Web30k with RankBCE loss.

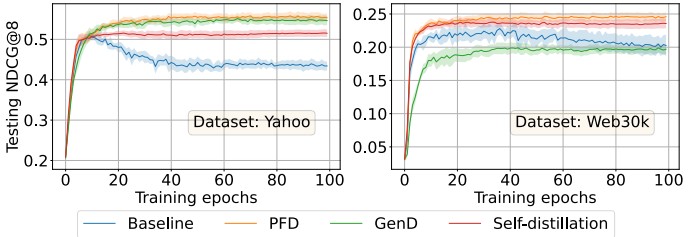

Figure 2: Testing NDCG@8 curve on Yahoo and Web30k with RankBCE loss. The baseline is no-distillation training with only regular features. It overfits in the later epochs. The performance of self-distillation and GenD depends on dataset. PFD has the best performance in both settings.

| Training method | NDCG @8 | NDCG @16 | NDCG @32 |
|---|---|---|---|
| | Loss function: RankBCE;  Dataset: Yahoo | | |
| No-distillation | $0.517 \pm 0.005$ (+0.0%) | $0.557 \pm 0.005$ (+0.0%) | $0.582 \pm 0.005$ (+0.0%) |
| Self-distillation | $0.522 \pm 0.004$ (+1.0%) | $0.559 \pm 0.003$ (+0.3%) | $0.585 \pm 0.003$ (+0.5%) |
| GenD | $0.557 \pm 0.005$ (+7.7%) | $0.583 \pm 0.005$ (+4.7%) | $0.607 \pm 0.004$ (+4.3%) |
| PFD | $\mathbf{0.566 \pm 0.004}$ **(+9.5%)** | $\mathbf{0.592 \pm 0.005}$ **(+6.2%)** | $\mathbf{0.614 \pm 0.003}$ **(+5.4%)** |
| | Loss function: RankBCE;  Dataset: Istella | | |
| No-distillation | $0.402 \pm 0.001$ (+0.0%) | $0.446 \pm 0.001$ (+0.0%) | $0.472 \pm 0.001$ (+0.0%) |
| Self-distillation | $0.402 \pm 0.001$ (+0.0%) | $0.446 \pm 0.002$ (+0.1%) | $0.472 \pm 0.001$ (+0.2%) |
| GenD | $0.380 \pm 0.001$ (-5.4%) | $0.426 \pm 0.001$ (-4.4%) | $0.455 \pm 0.001$ (-3.6%) |
| PFD | $\mathbf{0.417 \pm 0.002}$ **(+3.7%)** | $\mathbf{0.461 \pm 0.002}$ **(+3.4%)** | $\mathbf{0.486 \pm 0.002}$ **(+3.1%)** |
| | Loss function: RankBCE;  Dataset: Web30k | | |
| No-distillation | $0.241 \pm 0.006$ (+0.0%) | $0.267 \pm 0.006$ (+0.0%) | $0.301 \pm 0.006$ (+0.0%) |
| Self-distillation | $0.241 \pm 0.004$ (-0.0%) | $0.268 \pm 0.004$ (+0.5%) | $0.299 \pm 0.004$ (-0.5%) |
| GenD | $0.205 \pm 0.004$ (-15.0%) | $0.233 \pm 0.005$ (-12.7%) | $0.269 \pm 0.005$ (-10.6%) |
| PFD | $\mathbf{0.252 \pm 0.004}$ **(+4.5%)** | $\mathbf{0.281 \pm 0.006}$ **(+5.5%)** | $\mathbf{0.314 \pm 0.005}$ **(+4.3%)** |

Table 2: **Evaluation of PFD** and other related algorithms on Yahoo, Istella, Web30k with RankBCE loss function. We set $\alpha = 0.5$ (Equation (1)) for all evaluated settings. The baseline model is trained without privileged features. We also evaluate "self-distillation" [FLT$^+$18] and "GenD" [LPSBV16] for comparisons, see detailed model description Section 3.2. Experiments on Yahoo and Istella are repeated for 5 independent runs, and the offical 5 fold-splits are used for Web30k experiments. The best checkpoint (measured by testing NDCG@8) of 100 training epochs is used for evaluation, with mean and standard deviation of 5 runs reported. The results show that PFD has the best performance on all evaluated settings. See Table 4 for complete results with pairwise loss (RankNet) and teacher models' performance.

Table 2 shows that PFD has the best performance on all evaluated settings. We remark that (1) the only difference between PFD and self-distillation is that the teacher in PFD additionally uses privileged features and therefore has better prediction accuracy than the teacher in self-distillation. Comparing PFD with self-distillation reveals the improvement of using "privileged features" for distillation; (2) the performance of GenD is worse than no-distillation on Istella and Web30k. The reason for such inferior performance is that the teacher model in GenD only uses privileged features (and not regular features). For Istella and Web30k, only using privileged features is not sufficient to generate good predictions. The teachers in GenD are also worse than no-distillation, see Appendix A.4.

## 4.2 Ablation study on public datasets

**PFD is not sensitive to $\alpha$.** In former experiments, we kept the mixing ratio of teacher loss and data loss to be $\alpha = 0.5$. Here we evaluate the sensitivity of PFD to parameter $\alpha$. The experiments here use the Yahoo dataset and RankBCE loss. From the left-hand side of Figure 3, we see that PFD delivers good performance over a large range of $\alpha$.

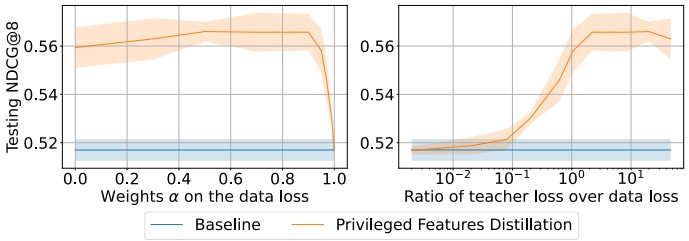

Figure 3: Sensitivity of privileged features distillation to $\alpha$.

However, it is worth noting that the teacher loss is typically much larger than the data loss (e.g., about 20 times larger in this set of experiments), since the teacher's predictions are much denser learning targets. The right-hand side plot of Figure 3 takes the scale of both losses into consideration. It shows that PFD yields the best performance only when the teacher loss dominates the distillation loss.

**PFD brings a larger gain when the positive labels are sparse.** Recall that we view negative query groups as unlabeled data. Here we evaluate the performance of PFD under different numbers of positive labels. Specifically, by reducing $\tau_{\text{target}}$ from 4.8 to 0.4, we can increase the percentage of

positive query groups (i.e., query groups with at least one $y = 1$). The relative improvement over baseline is shown in Figure 4. While it is known that distillation works better when there are more unlabeled samples, Figure 4 shows that PFD further amplifies such gains: the relative gain of PFD over self-distillation also increases as the positive labels become sparser. Such benefit is especially favorable in recommendation systems, where the positive labels (e.g., click) are naturally very sparse.

**Correlation between privileged features and target.** It is believed that privileged features that are discriminative (e.g., high correlation with the target) lead to accurate teacher predictions, and thus benefit the distillation [XLG+20]. However, we show that PFD has poor performance when the privileged features are too discriminative.

Specifically, we modify the experiment setting such that all the features in the datasets are used as regular features, while the privileged features $z$ are generated according to $z = \mathbb{I}(t \cdot r + G_1 > t \cdot \tau_{\text{privileged}} + G_0)$, where $G_1$ and $G_0$ have the same values as in binary label $y$ generation (Equation (2)). By changing $\tau_{\text{privileged}}$, we can obtain privileged features $z$ with different correlations with the label $y$. For instance, when $\tau_{\text{privileged}} = \tau_{\text{target}}$, then $z$ can perfectly predict $y$ (since $z = y$ by definition); and $z$ becomes less discriminative when $\tau_{\text{privileged}}$ gets smaller. Using $z$ as the privileged feature, we have the PFD results in Figure 5.

Notice that the privileged feature with the largest correlation with $y$ does not give the best performance. We believe the reason is that as the correlation of $z$ and $y$ increases, the privileged feature becomes so "discriminative" that it can explain almost all the variance in $y$, even the noise. As a result, teacher predictions have high variance, which leads to high-variance student estimates and inferior testing performance. See Section 5.2 for theoretical insights.

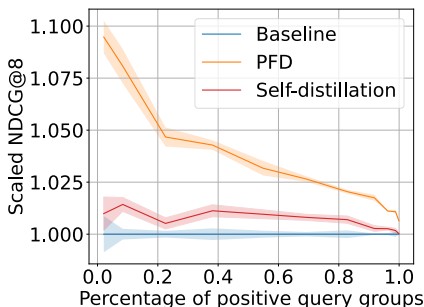

Figure 4: Distillation performance under different percentages of positive query groups. All testing NDCG@8 scores are scaled w.r.t. the baseline performance. PFD yields larger improvement when the labeled samples are scarce.

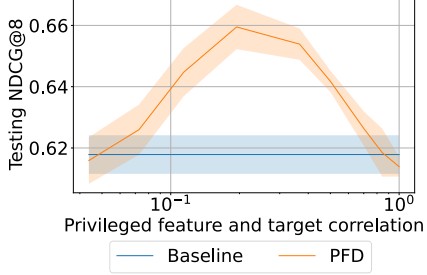

Figure 5: The performance of PFD using privileged features with different correlations with $y$. The privileged features with the highest correlation with $y$ do not yield the best distillation performance.

### 4.3 Evaluation on Amazon's dataset

**Dataset overview and ranking model.** The dataset is derived from Amazon's logs which contains *query* and *product title* text, the *position* at which the product was shown, and the user's behaviors *click, add-to-cart,* and *purchase*. The ranking model is a multi-layer transformer that maps query and product title to an estimate of the purchase likelihood. The goal is to rank the products that are more likely to be purchased first.

**Efficacy of PFD.** Here we evaluate the performance of PFD. Notice that the position is a privileged feature as it is not available as an input during online serving (it is the output of the ranking model that becomes position). Further, the click and add-to-cart are naturally privileged features, since one cannot know which of the product will be clicked or added to cart before showing the products.

The baseline no-distillation model only takes query and product title as input, while the teacher models in PFD additionally take positions or clicks or add-to-cart as privileged features. As in public datasets, we only use the positive query groups to train the teacher model and use all query groups for distillation. We additionally use pretraining then finetuning as another baseline, as predicting "click" or "add-to-cart" can serve as pretraining tasks. The experiment results are shown in Table 3.

**Extension: Multi-teacher distillation.** Inspired by [FSK+17, ZXHL18], we also evaluate the multi-teacher distillation, where the student learns from more than one teacher. We adopt three privileged teachers which take positions, clicks, and add-to-cart as input, respectively. We calculate the loss w.r.t. each of the teachers' predictions and use the average as "teacher loss" in Equation (1). Intuitively, the student model is trained to learn from an "ensemble" of teacher models. The multi-teacher PFD yields the best performance, an 11.2% improvement on testing NDCG@8 over the baseline model.

| Models | | | Purchase NDCG | | |
|---|---|---|---|---|---|
| | | | @8 | @16 | @32 |
| Baseline | No-distillation | | 1.000 | 1.000 | 1.000 |
| Pretrain + finetune | Pretrain on click | | 1.043 | 1.035 | 1.031 |
| | Pretrain on click + finetune on purchase | | 1.077 | 1.060 | 1.054 |
| | Pretrain on add | | 1.058 | 1.046 | 1.042 |
| | Pretrain on add + finetune on purchase | | 1.059 | 1.045 | 1.041 |
| Distillation | No privileged feature | Teacher | 1.000 | 1.000 | 1.000 |
| | | Student | 1.028 | 1.021 | 1.019 |
| Privileged feature distillation | position as privileged feature | Teacher | 1.182 | 1.135 | 1.121 |
| | | Student | 1.078 | 1.057 | 1.051 |
| | click as privileged feature | Teacher | 2.121 | 1.902 | 1.827 |
| | | Student | **1.082** | **1.064** | **1.057** |
| | add as privileged feature | Teacher | 2.292 | 2.053 | 1.972 |
| | | Student | 1.073 | 1.056 | 1.051 |
| Multi-teacher distillation | Distill from position + click + add | | **1.112** | **1.085** | **1.077** |

Table 3: **Evaluation of PFD on Amazon's dataset.** We use position (i.e., the position at which the product was shown in the logged query group), or click, or add-to-cart as privileged features. The baseline model takes the text of "query" and "product title" as inputs. All reported results are scaled w.r.t. the performance of baseline. The results demonstrate that (1) PFD is significantly better than baseline and pretraining-then-finetuning; (2) using add-to-cart as a privileged feature gives the best teacher model, which, however, does not lead to the best distillation result. This observation agrees with the result from Figure 5 where the most discriminative privileged feature does not give the best distillation performance; (3) learning from multiple teachers (i.e., "teacher loss" in Equation (1) being the averaged loss w.r.t. each of the teachers' predictions) further improves the PFD performance and leads to the best ranking performance.

## 5 Theoretical Insights

In this section, we present theoretical insights on why and when PFD works via analysis on linear models. While our empirical focus is on ranking problems, our theoretical insights are more general. Consider the following learning problem: the regular feature $\mathbf{x} \in \mathbb{R}^{d_x}$ is drawn from a spherical Gaussian distribution $\mathcal{N}(0, \mathbb{I}_{d_x})$ and an un-observable feature $\mathbf{u} \in \mathbb{R}^{d_u}$ is drawn from $\mathcal{N}(0, \mathbb{I}_{d_u})$. With two unknown parameters $\mathbf{w}^* \in \mathbb{R}^{d_x}$ and $\mathbf{v}^* \in \mathbb{R}^{d_u}$, the label $y$ is generated as following:

$$y = \mathbf{x}^\top \mathbf{w}^* + \mathbf{u}^\top \mathbf{v}^* + \epsilon, \quad \epsilon \sim \mathcal{N}(0, \sigma^2), \tag{3}$$

where $\epsilon$ represents the label noise. During training time, we observe the features $\mathbf{z} = \mathbf{u}$ as privileged features. Suppose that the labeled training set $\mathcal{S}_{label} = \left\{ \mathbf{X} \in \mathbb{R}^{n \times d_x}, \mathbf{Z} \in \mathbb{R}^{n \times d_z}, \mathbf{y} \in \mathbb{R}^n \right\}$ and the unlabeled set $\mathcal{S}_{unlabel} = \left\{ \mathbf{X}_{(u)} \in \mathbb{R}^{m \times d_x}, \mathbf{Z}_{(u)} \in \mathbb{R}^{m \times d_z} \right\}$ are generated according to the afore-mentioned data generation scheme. Let $\mathbf{X}_{(a)} = [\mathbf{X}; \mathbf{X}_{(u)}] \in \mathbb{R}^{(n+m) \times d_x}$ and $\mathbf{Z}_{(a)} = [\mathbf{Z}; \mathbf{Z}_{(u)}] \in \mathbb{R}^{(n+m) \times d_z}$ be the all the inputs from both labeled and unlabeled datasets. The goal is to learn to predict $y$ with only regular feature $\mathbf{x}$ as input.

### 5.1 PFD works by reducing estimation variance

Let $\widehat{\mathbf{w}}_{reg}$ denote the model learned by standard linear regression and $\widehat{\mathbf{w}}_{pri}$ be the model learned by privileged features distillation. For simplicity, we consider the case with $\alpha = 0$, i.e., only learning from the teacher's prediction during distillation. Specifically, the standard linear regression only uses the set $\mathcal{S}_{label}$, and $\widehat{\mathbf{w}}_{reg}$ is obtained by regressing $\mathbf{y}$ on $\mathbf{X}$. PFD, on the other hand, first uses $\mathcal{S}_{label}$ to regress $\mathbf{y}$ on $[\mathbf{X}; \mathbf{Z}]$. The learned model is then used to generate predictions $\widehat{\mathbf{y}}$ for $\mathcal{S}_{label} \cup \mathcal{S}_{unlabel}$. Finally, $\widehat{\mathbf{y}}$ is regressed on $\mathbf{X}_{(a)}$, which gives $\widehat{\mathbf{w}}_{pri}$. We have the following result on the merit of PFD:

**Theorem 1.** *For standard linear regression, we have that*

$$\mathbb{E}_{\mathbf{X},\mathbf{y}}\|\mathbf{w}^* - \widehat{\mathbf{w}}_{reg}\|_2^2 = O\left(\frac{d_x \cdot (\sigma^2 + \|\mathbf{v}^*\|^2)}{n}\right).$$

*For privileged features distillation, we have that*

$$\mathbb{E}_{\mathbf{X}_{(a)},\mathbf{Z}_{(a)},\mathbf{y}}\|\mathbf{w}^* - \widehat{\mathbf{w}}_{pri}\|_2^2 = O\left(\frac{d_x \cdot \sigma^2}{n}\right) + O\left(\frac{d_x \cdot \|\mathbf{v}^*\|^2}{n+m}\right) + O\left(\frac{1}{n \cdot m}\right).$$

Notice that $\text{var}(y|\mathbf{x}) = \sigma^2 + \|\mathbf{v}^*\|_2^2$, where $\sigma^2$ corresponds to the label noise and $\|\mathbf{v}^*\|_2^2$ corresponds to the variance that can be explained by the privileged features. The result shows that PFD can explain a proper part of the variance in $y$ by privileged features $\mathbf{z}$. By learning from the teacher's predictions, PFD can therefore reduce the variance of $\widehat{\mathbf{w}}_{pri}$ by exploiting the privileged features and the unlabeled samples. On the other hand, when learning with plain linear regression, the label variance corresponding to $\mathbf{z}$ is treated as noise, which leads to estimation with higher variance.

**Remark 2.** ***Why GenD has worse-than-baseline performance.*** *Notice that the teacher model in GenD uses privileged features only. GenD has inferior performance for two reasons: (1) the privileged features alone are not enough for the teacher model to generate good predictions; and (2) when $\mathbf{z}$ is independent of $\mathbf{x}$, the predictions from the GenD's teacher are not learnable for the student.*

### 5.2 PFD has inferior performance when the privileged features are too discriminative

To understand the performance of PFD under different privileged features, consider the setting where $\mathbf{z} \in \mathbb{R}^{d_z}$ is the first $d_z$ coordinates of $\mathbf{u}$. When $d_z = d_u$, it recovers the setting in previous subsection. Notice that the larger $d_z$ becomes, the better $(\mathbf{x}; \mathbf{z})$ can predict $y$. While one might expect that $d_z = d_u$ (i.e., when the privileged features contain the most information about $y$) leads to the best distillation performance, our next result shows that such belief is not true in general. Let $\mathbf{v}_{\mathbf{z}}^*$ be the part of $\mathbf{v}^*$ that corresponds to $\mathbf{z}$ (i.e., the first $d_z$ coordinates of $\mathbf{v}^*$), we have:

**Theorem 2.** *For privileged features distillation, we have that*

$$\mathbb{E}_{\mathbf{X}_{(a)},\mathbf{Z}_{(a)},\mathbf{y}}\|\mathbf{w}^* - \widehat{\mathbf{w}}_{pri}\|_2^2 = \frac{d_x \cdot (\sigma^2 + \|\mathbf{v}^*\|_2^2 - \|\mathbf{v}_{\mathbf{z}}^*\|_2^2)}{n - d_x - d_z - 1} + \frac{d_x \cdot \|\mathbf{v}_{\mathbf{z}}^*\|_2^2}{n + m - d_x - 1} + O\left(\frac{1}{n \cdot m}\right).$$

As we increase $d_z$ from 0 to $d_u$, $\|\mathbf{v}_{\mathbf{z}}^*\|$ also increases. The teacher, therefore, explains more variance in $y$ and contributes to a smaller error in the student estimate $\widehat{\mathbf{w}}_{pri}$. However, the denominator of the first term decreases as $d_z$ increases, which leads to a higher variance (thus less accurate) student parameter estimate $\widehat{\mathbf{w}}_{pri}$. Combining the two effects, the privileged features $\mathbf{z}$ that contain the most information about $y$ do not yield the best distillation performance. This matches the non-monotone observation in Figure 5; and the results in Table 3 where using add-to-cart (i.e., the most informative feature for predicting purchase) does not give the best PFD result.

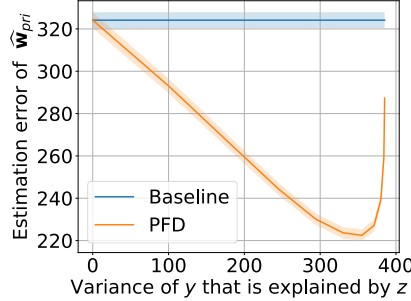

Figure 6: PFD with different sets of privileged features. The most predictive $\mathbf{z}$ does not give the best PFD performance.

**Example 1.** *Consider the data generation as shown in Equation (3). We set $d_x = 10, d_u = 10, n = 30, m = 200$, and draw $\mathbf{w}^*$ from a spherical Gaussian distribution $\mathcal{N}(0, \mathbb{I}_{d_x})$. Further, we set $\sigma = 15$, and let $\mathbf{v}^* = [10, 9, \cdots, 2, 1]$. We evaluate the performance of the standard linear regression and the privileged features distillation with $d_z$ from 0 to 10. The results in Figure 6 shows that the most predictive $\mathbf{z}$ does not give the best PFD performance.*

## 6 Conclusion

In this paper, we take a step toward understanding PFD in learning-to-rank. We first evaluate PFD on three public ranking datasets (Yahoo, Istella, and MSLR-Web30k) and an industrial-scale ranking problem derived from Amazon's search logs. Our evaluation shows that PFD has the best

performance in all evaluated settings. We further conduct comprehensive empirical ablation studies, which demonstrates the efficacy and robustness of PFD and uncovers an interesting non-monotone behavior – as the predictive power of privileged features increase, the performance of PFD first increases and then decreases. Finally, we present theoretical insights for PFD via rigorous analysis for linear models. The theoretical results show that (1) PFD is effective by reducing the variance of student estimation; and (2) a too predictive privileged teacher produces high variance predictions, which lead to high variance (less accurate) student estimates and inferior testing performance.

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
