# OpenReview forum: "Toward Understanding Privileged Features Distillation in Learning-to-Rank"
_NeurIPS.cc/2022/Conference — NeurIPS 2022 Accept_

### Official Review · Reviewer_XfRZ · 2022-07-07

**Rating:** 6
**Confidence:** 3
**Soundness:** 3 good
**Presentation:** 4 excellent
**Contribution:** 2 fair

**Summary:**

This paper studies the privileged feature distillation (PFD) problem. The paper consists of two parts - empirical evaluation on public datasets and an industry dataset, followed by some theoretical analysis on linear models. The paper focuses on understanding an existing method instead of proposing new methods. The empirical part confirms that PFD is effective on several datasets. Some ablations are provided in terms of label sparsity, etc. On the three public datasets, the setting is controlled - binary labels are generated and privileged features are manually selected. The evaluation on the industry dataset looks more standard. On the theoretical part, the analysis is done on linear models. The insights found include 1) PFD works by reducing estimation variance. 2) Why too discriminative privileged features can hurt.

Overall the reviewer finds this paper well written in general. The reviewer feels the empirical study meets the bar by performing on multiple datasets and comparing with sensible baselines. The theoretical part looks reasonable but not surprising.


**Questions:**

Please see above.

One comment is that the reviewer does not find anything specific to learning to rank except for the dataset used. The reviewer does not feel it is a big issue but is a bit surprised. The authors may consider clarifying why learning to rank is the topic, as in the title and such.

**Strengths And Weaknesses:**

Strength
The reviewer has personal interest in the topic (though not sure about the interest from a wider group). The paper is generally well written.
The reviewer feels the empirical evaluation meets the bar, by performing on multiple datasets and comparing with sensible baselines. Some ablations look interesting.
The theoretical part is clear and focus on important aspects.

Weakness
The theoretical analysis is not particularly deep. The conclusions are intuitive and nothing is surprising. Focusing on linear models is ok but may not be very impressive.
The controlled setting on public datasets seem a bit artificial and may bias towards the concerned methods. For example, the most correlated features are used as privileged features. Considering other options could be more comprehensive.
The paper does not show any online experiments, which is the major motivation of PFD.

---

> ### Author Response · Authors · 2022-08-02
> **Response to Reviewer XfRZ**
>
> Thank you for appreciating our work and the suggestions! For questions and concerns raised in the review:
>
> **[Limited theoretical analysis]** Our theoretical analysis of linear models provides interesting and non-trivial insights into PFD:
> - The theoretical analysis explains an interesting non-monotone behavior observed for deep models - i.e, the most predictive privileged features do not give the best distillation result (see Fig 5 and 6). Such a phenomenon was not previously expected or explained.
> - Our analysis shows that PFD still improves the student’s performance when the privileged features are independent of regular features (i.e., nothing in the privileged features can be predicted by the regular features), which is interesting and not previously realized. See discussion in Remark 1 and Section 5.
> - We prove that the PFD is effective by reducing the learning target variance.
>
> We kindly emphasize that it is generally acknowledged that fine-grained theoretical analysis (of the kind we do in our paper) for deep models is challenging, and often involves its own (often unrealistic) assumptions. So we took a two-pronged approach in our paper:
> 1. Our linear analysis, despite its technical simplicity, brings clear insights into PFD and explains various observations for deep learning models.
> 2. Ablation studies and principled comparison demonstrate the power of PFD on a wide set of scenarios for larger neural models and match our theoretical results.
>
>
> **[The choice of privileged features]** Besides the synthetic experiments that use the most correlated features as privileged ones, there are two other types of privileged features empirically evaluated: (1) the binary synthetic privileged features (see Lines 239 - 263) used for evaluating the correlation between synthetic features and target; and (2) real privileged features (Section 4.3) where “position”, “click”, or “add-to-cart” are all evaluated for PFD. The empirical evaluations demonstrate that the PFD scheme is beneficial for a large variety of privileged features.
>
> To further demonstrate the effectiveness, we additionally run experiments with different numbers of features used as privileged features, and the results demonstrate that PFD brings consistent improvement over different numbers of privileged features. Specifically, comparing the NDCG@8 metric with the baseline, we have the following results
> - Yahoo: using 100 features as privileged features gives a 0.9% improvement; using 200 privileged features gives a 9.5% improvement (this is reported in the paper); using 300 privileged features gives a 13.1% improvement over baseline.
> - Istella: using 30 features as privileged features gives a 2.9% improvement; using 40 privileged features gives a 3.7% improvement (this is reported in the paper); using 100 privileged features gives a 5.9% improvement.
> - MSLRWEB30k: using 20 features as privileged features gives a 2.0% improvement; using 40 privileged features gives a 4.5% improvement (this is reported in the paper); using 60 privileged features gives a 5.7% improvement.
>
> Comparing the results, it shows that PFD can deliver a consistent improvement over baseline with a wide range of privileged features.
>
>
> **[Online experiments]** For “online” experiments, the PFD is evaluated on an industrial-level proprietary dataset, where the PFD teacher uses “click”, or “add-to-cart” features that can only be used on offline data. The PFD student only relies on “query keywords” and “product title”, which can be deployed online directly.
>
> **[Clarification for LTR]** This paper focuses on LTR because the privileged features naturally arise from LTR problems and recommendation systems (e.g., the “click” and “add-to-cart” are naturally privileged features for “purchase”). We agree with the reviewer that the PFD method itself (and our understanding of PFD) can be applied much more broadly. We will add clarification in our next revision.
>
> We hope our response and additional experiments have addressed all of your concerns. Thank you again!

---

### Official Review · Reviewer_iukh · 2022-07-07

**Rating:** 7
**Confidence:** 4
**Soundness:** 3 good
**Presentation:** 3 good
**Contribution:** 2 fair

**Summary:**

The authors provide an empirical study of Privileged Features Distillation (PFD) for Learning-to-Rank problems (LTR), applied on 3 public datasets and one private industrial dataset (Amazon search logs).
The principle of PFD is based on two models: 1) one, which learns with all the features available (including the privileged ones) and will play the role of a « teacher » to a 2) second model, the « student » model which is trained using only the regular features and into which teacher information is transferred via distillation.
PFD is compared against 4 other baselines:
- no distillation (training only on regular features, no teacher, only one model)
- pre-training on privileged features followed by fine-tuning with only regular features
- self-distillation (the teacher model is trained only on non-privileged features)
- and generalized distillation (GenD, the teacher model is trained only on privileged features)
Experiments show PFD performs better or as good as the baselines.
An ablation study and theoretical analysis focused on linear models finally help to understand when and why PFD works.

**Questions:**

Q1: Equation 2:
First comment: By reading the sequel we understand why we need the temperature-based schema to transform human-annotated relevance scores into binary relevance (and not a simple threshold rule): the objective is to have a non-trivial artificial relationship between label and privileged features for the purpose of some demonstration later but this could be clarified before.

How \tau target is defined?

Q2: Regarding the sensitivity trend to \alpha, do we observe the same for other datasets?

Q3: Evaluation on Amazon dataset. How is handled the feature of « product title » in the model? What does the « query » feature look like?

Q4: Table 3: multi-teacher distillation requires 3 teacher models, each of them trained on all regular features +  one privileged feature at a time. What about one single teacher (and one teacher loss) with all the privileged features at the same time? What is the corresponding performance?


**Limitations:**

Yes, limitations are enunciated. This is even the purpose of the paper to determine in which cases PFD performs or does not perform.
Potential negative societal impact of the work is not mentioned as this is more a generic LTR problem.



**Strengths And Weaknesses:**

Strengths:

S1 (clarity, quality): The paper is well-written and its clarity makes it easy to follow and enjoyable to read.

S2 (significance): This paper is a first attempt to bridge the gap of lack of performance understanding of PFD.
The extensive experiments and theoretical analysis conducted in this paper help to better understand PFD and support some intuitions (ex: PFD cannot do miracles if the most discriminative features for the task at hand are privileged - this shows the superiority of PFD over GenD) and less intuitive results (teacher loss should dominate distillation loss, PFD works better with sparser labels, PFD reduces estimation variance).
Supplemental work on the use of a teacher model with imputed privileged features during inference is also interesting.


Weaknesses:

W1 (clarity, minor): This is minor but as the concept of privileged feature is not limited to learning-to-rank problems, the first sentence of the abstract can be misleading.

Typos:
-row 54: « PDF »
-row 178: indicator function mentioned for the first time, always helpful for the reader to name it (and define it)!

---

> ### Author Response · Authors · 2022-08-02
> **Response to Reviewer iukh**
>
> Thank you for appreciating our work and the suggestions! For the questions and concerns raised in the review:
>
> **[Clarification for LTR]** This paper focuses on LTR because the privileged features naturally arise from LTR problems and recommendation systems (e.g., the “click” and “add-to-cart” are naturally privileged features for “purchase”). We agree with the reviewer that the PFD method itself (and our understanding of PFD) can be applied much more broadly. We will add clarification in our next revision.
>
> **[Definition of $\tau_{target}$]** $\tau_{target}$ is a hyper-parameter that controls the sparsity of the artificial target. It is set to 4.8 for all the experiments (see line 181), except for the ablation study on label sparsity, where $\tau_{target}$ ranges from 4.8 to 0.4.
>
> **[Sensitivity to $\alpha$ in other datasets]** As requested in the review, we also evaluated the alpha impact on Istella and MSLRWEB30k, and the results demonstrate the same conclusion as in the paper - the PFD is not sensitive to alpha on both datasets. Specifically, we evaluated a list of alpha: [0.3, 0.5, 0.7, 0.9, 0.95, 0.99, 0.999] and the results on Istella show that for all $\alpha \le 0.95 $, PFD can deliver an over 2.5% improvement in NDCG@8; for MSLRWEB30k, for all $\alpha \le 0.95$, PFD can stably deliver an over 2.6% improvement. We will include this additional set of results in our revision.
>
> **[Amazon dataset features]** The “product title” and “query” are all represented in raw text. We adopt standard transformer models to extract numerical features from them and make purchase predictions.
>
> **[Using position+click+add in one teacher]** This is a very good question, and we evaluated a teacher with position + click + add as privileged features in our preliminary experiments. The results show that using position+click+add in one teacher has a very similar performance as using “add-to-cart” as the only privileged feature. The model heavily relies on “add-to-cart” to make purchase predictions. Such an observation is also intuitive - since the causal relation is “click” -> “add-to-cart” -> “purchase”, by conditioning on “add-to-cart”, the “click” has no information on “purchase”. We will evaluate this teacher model again and include the results in our revision.

---

> > ### Comment · Reviewer_iukh · 2022-08-09
> > **Thanks for your feedback!**
> >
> > Thanks for your feedback! It makes sense to me.
> > I keep my current score.

---

### Official Review · Reviewer_GHC5 · 2022-07-11

**Rating:** 5
**Confidence:** 3
**Soundness:** 3 good
**Presentation:** 2 fair
**Contribution:** 3 good

**Summary:**

In this work, the authors studied about the Privileged Features Distillation (PFD), where there are indicative features available in training but missing in serving, so as to leverage these privileged features, distillation of a teacher model trained with privileged features is deployed. In specific, in the PFD proposed in the work, the teacher model leverages both privileged features and regular features available in serving. The proposed setting is shown to be better than all baselines on the public and industrial datasets. The authors also provided emprical explaination on why and when PFD works by ablation study and theory on linear models. The main contributions mainly include a practically applicable method PFD and the theoretical understanding of the method.

**Questions:**

One puzzle to me is in the Appendix Table 4, I find some PFD students perform as good as, sometimes even better than their PFD teachers (on Istella and Web30k). How could the students without the most explainable privileged features outperform the teachers with both regular and privileged features? Because the labelled dataset is much smaller? It would be helpful if the authors could explain this.

How are the correlation of selected privileged features? In the synthetic data, the authors picked the most correlated features from each public dataset as the privileged features. On the other hand, the authors showed that there is a non-monotonic dependence of student model performance and the correlation of the privileged features. It sounds like selecting the most correlated features are not optimal except they can show the correlations of the most correlated features are arround the range of the optimal performance.

**Limitations:**

N.A.

**Strengths And Weaknesses:**

Strengths
1. The paper is overall well-written, with all main contributions listed and sufficient ablation study.
2. First work gives reasonable understanding of why using privileged feature distillation works.

Weaknesses
1. Figure layouts look a bit wierd. Authors may have used some special template. Usually, a figure will take a full column instead of just a floating panel.
2. Some notation might be better explained in the main text, for example, RankBCE is short for binary cross entropy loss. It's not very interpretable without checking the appendix.

---

> ### Author Response · Authors · 2022-08-02
> **Response to Reviewer GHC5**
>
> Thank you for your comments and suggestions! We will adjust the figure layouts and clarify the notation in our revision. For other questions and concerns raised in the review:
>
> **[Student outperforming teacher model in Appendix Table 4]** There are 3 reasons contributing to this observation:
> 1. Notice that the differences between those PFD students and teachers are relatively small - they mostly fall within 1 standard deviation. Their performance difference could be the result of randomness.
> 2. It is more common for Web30k to have similar PFD student and PFD teacher performances. We believe the reason is that the privileged features on Web30k are not strong enough - note that among the 3 datasets, the PFD teacher in Web30k has the least improvement over baseline. Therefore, it is less surprising that the PFD student can perform similarly to the PFD teacher on Web30k.
> 3. The distillation scheme in general brings some performance improvement to the student model, regardless of privileged features - this is observed even when the teacher model and student model use the same model and features (see the line of work on “self-distillation”). This additionally contributes to a better performance of the student model.
>
> **[Correlation of selected privileged features]** For Yahoo, the selected privileged features have correlations ranging from 0.453 to 0.302; for Istella, the correlations are from 0.257 to 0.159; for Web30k, the correlations are from 0.235 to 0.148.
>
> While those correlations are in the optimal regime suggested by Figure 5, we would like to emphasize that the optimal correlation depends on the regular features and the target, which is different for different datasets. The main message of Figure 5 is the interesting non-monotone behavior that is not previously reported. It does not indicate the optimal target and privileged feature correlation in general.
>
> We hope our response has addressed all of your concerns, and kindly hope the score can be re-evaluated accordingly. Thank you again!

---

### Official Review · Reviewer_76AW · 2022-07-13

**Rating:** 5
**Confidence:** 4
**Soundness:** 2 fair
**Presentation:** 3 good
**Contribution:** 2 fair

**Summary:**

The paper empirically evaluates distillation approach for privileged features. Teacher is trained with privileged features that are not available during inference, and student then aims to replicate performance of the teacher without these features. Empirical results on public and proprietary datasets show that this approach achieves better performance than baselines. Authors further analyse theoretical properties of this distillation approach in the case of linear models and show that it has desirable properties.

**Questions:**

-On line 185 why these specific numbers of features were chosen: 200, 50, and 40? What happens when you use more/less features for each dataset? I would think that this choice has significant effect on performance.
-Why "Pretrain on click" and "Pretrain on add" performs better than baseline? These models are not finetuned on purchases, does this mean that clicks and adds are better training targets?
-Have you tried to use position+click+add in one teacher? Seems odd to only use one of them when a much better teacher can probably be obtained with all three.

**Strengths And Weaknesses:**

Strengths
Paper is well written and easy to follow. Empirical evaluation is quite thorough and I particularly enjoyed the section on the Amazon dataset although these results are not reproducible. Theoretical results provide some insight into properties of PFD and could be used as a stepping stone for more analysis.

Weaknesses
I find that the paper has limited novelty. Section 4 is all about empirical evaluation and while it has useful insights the novelty limited.  Theoretical analysis in Section 5 is probably the most novel part but it only analyses linear models, and is of limited utility for the complex gradient boosting or deep learning models that are typically used for ranking. Moreover, in most cases it should be possible to use privileged features directly as additional targets. This is cheaper than distillation since it doesn't require training teacher models. Pretrain + finetune results in Table 3 perform similarly to distillation with one teacher, and call into question whether distillation is necessary here at all. I suspect that by carefully tuning weights in a multi-target loss it should be possible to recover multi-teacher performance with just one model.

---

> ### Author Response · Authors · 2022-08-02
> **Response to Reviewer 76AW - PART 2 / 2**
>
> (Please see the first half of the response in PART 1 / 2)
>
> **[Questions on pretraining on click and add-to-cart outperforming baseline without finetuning]** Indeed, this is an intriguing phenomenon, and has the following explanation: in practice, it is observed that clicks and adds-to-cart are heavily correlated with purchases, while also being an order of magnitude more frequent in datasets. The extreme sparsity of purchase makes it very challenging to learn from purchase directly. Clicks and adds, on the other hand, provide many more samples that benefit the model training, albeit of a slightly different but correlated target. The models pretrained on click or add-to-cart are not well calibrated for predicting purchase probability, but calibration matters less if the final objective is sorting (as is the case in ranking).
>
> **[Using position+click+add in one teacher]** This is a very good question, and we evaluated a teacher with position + click + add as privileged features in our preliminary experiments. The results show that using position+click+add in one teacher has a very similar performance as using “add-to-cart” as the only privileged feature. The model heavily relies on “add-to-cart” to make purchase predictions. Such an observation is also intuitive - since the causal relation is “click” -> “add-to-cart” -> “purchase”, by conditioning on “add-to-cart”, the “click” has no information on “purchase”. We will evaluate this teacher model again and include the results in our revision.
>
> We hope our response and additional experiments have addressed all of your concerns, and kindly hope the score can be re-evaluated accordingly. Thank you again!

---

> ### Author Response · Authors · 2022-08-02
> **Response to Reviewer 76AW - PART 1 / 2**
>
> Thank you for the detailed comments and suggestions! For the concerns raised in the review:
>
> **[Limited novelty]** While we found out that the notion of Privileged Features Distillation did exist previously, it was demonstrated as a one-off idea on a private dataset, with limited comparative baselines and no analysis providing intuition/rigor. The main novelty of our paper is providing a theoretical and empirical understanding as to why and when PFD works and when it does not. For example,
> 1. we explain the reason for why the usefulness of a privileged feature is non-monotone in its predictive power, which we observed in practice but was not previously reported,
> 2. we show why PFD is more effective than other baseline approaches (like treating the privileged feature as an ancillary target) by rigorously analyzing scenarios like the one where the privileged feature is independent of the regular features (while still being predictive of the target). In such a scenario, the ancillary target approach will not work, but PFD does.
>
> Regarding the analysis being limited to linear models, it is generally acknowledged that fine-grained theoretical analysis (of the kind we do in our paper) for deep models is challenging and often involves its own (often unrealistic) assumptions. So we took a two-pronged approach in our paper:
> 1. Our linear analysis, despite its technical simplicity, brings clear insights into PFD and explains various observations for deep learning models.
> 2. Ablation studies and principled comparison demonstrate the power of PFD on a wide set of scenarios for larger neural models and match our theoretical results.
>
>
> **[Using privileged features as targets]** Indeed this is a natural question. While this approach has been known to often show gains over a baseline where no ancillary targets are used, its applicability is more limited than that of PFD. To see this, we draw attention to Remark 1 and Section 5 of our paper, where we consider the case where the privileged feature is independent of the regular features (while still remaining predictive of the target). In such a setting, using privileged features as an ancillary target (as the reviewer suggests) will provide no benefits because it is akin to predicting noise as far as only the regular features are concerned; however we show that PFD manages to capture and transfer the benefits of such an independent feature.
>
> **[Question on the choice of privileged features]** Yahoo, Istella, and Web30k have 700, 220, and 136 features in total, respectively. We use 25%~30% of the features as the privileged features, and that gives the choice of 200, 50, and 40 privilege features. Notice that we also constructed synthetic privileged features for ablation studies (see lines 239 - 263), which shows that PFD is effective over a wide range of privileged feature settings.
>
> As per the reviewer’s request, we additionally run experiments with different numbers of features used as privileged features, and the results demonstrate that PFD brings consistent improvement over different numbers of privileged features. Specifically, comparing the NDCG@8 metric with the baseline, we have the following results:
>
> - Yahoo: using 100 features as privileged features gives a 0.9% improvement; using 200 privileged features gives a 9.3% improvement (this is reported in the paper); using 300 privileged features gives a 13.1% improvement over baseline.
>
> - Istella: using 30 features as privileged features gives a 2.9% improvement; using 40 privileged features gives a 3.7% improvement (this is reported in the paper); using 100 privileged features gives a 5.9% improvement.
>
> - MSLRWEB30k: using 20 features as privileged features gives a 2.0% improvement; using 40 privileged features gives a 4.5% improvement (this is reported in the paper); using 60 privileged features gives a 5.7% improvement.
>
> Comparing the results, it shows that the choice of privileged features indeed has an impact on the improvement as suggested in the review; meanwhile, PFD can deliver a consistent improvement over baseline with a wide range of privileged features.
>
> (Please see the second half of the response in PART 2 / 2)

---

### Meta-Review · Area_Chair_6pwE · 2022-08-26

**Recommendation:** Accept
**Confidence:** Less certain

**Metareview:**

There is a consensus that the insights on the distillation of privileged information presented in the paper are interesting (e.g., possibility of distillation even if the privileged information is independent from x, non-monotonicity of the impact of privileged information vs correlation with the target feature), which is why the paper is recommended for acceptance.

Note that even after the rebuttal, several of the main weaknesses remain,

- it is not clear why the paper focuses "learning to rank" (apart from the original motivation of the authors), since the claims seem to hold as well in classification or regression
- the value of the theoretical analysis is limited, because it seems the authors considered the easiest setup where the phenomena illustrated in the experiment could be proved. In particular, they study linear least-square regression, which doesn't match any of their experiments.
- no novelty in terms of methods

In the end, the paper is borderline on the side of acceptance because the insights are significant enough.

**Award:**

No

---

### Decision · Program_Chairs · 2022-09-14

Accept